# Fast and Accurate Inference with Adaptive Ensemble Prediction for Deep Networks

## Abstract

Ensembling multiple predictions is a widely-used technique to improve the accuracy of various machine learning tasks. In image classification tasks, for example, averaging the predictions for multiple patches extracted from the input image significantly improves accuracy. Using multiple networks trained independently to make predictions improves accuracy further. One obvious drawback of the ensembling technique is its higher execution cost during inference.This higher cost limits the real-world use of ensembling. In this paper, we first describe our insights on relationship between the probability of the prediction and the effect of ensembling with current deep neural networks; ensembling does not help mispredictions for inputs predicted with a high probability, i.e. the output from the softmax. This finding motivates us to develop a new technique called adaptive ensemble prediction, which achieves the benefits of ensembling with much smaller additional execution costs. Hence, we calculate the confidence level of the prediction for each input from the probabilities of the local predictions during the ensembling computation. If the prediction for an input reaches a high enough probability on the basis of the confidence level, we stop ensembling for this input to avoid wasting computation power. We evaluated the adaptive ensembling by using various datasets and showed that it reduces the computation cost significantly while achieving similar accuracy to the naive ensembling. We also showed that our statistically rigorous confidence-level-based termination condition reduces the burden of the task-dependent parameter tuning compared to the naive termination based on the pre-defined threshold in addition to yielding a better accuracy with the same cost.

## 1 Introduction

The huge computation power of today's computing systems, equipped with GPUs, special ASICs, FPGAs, or multi-core CPUs, makes it possible to train deep networks using tremendous datasets. Although such high-performance systems can be used for training, actual inference in the real world may be executed on small devices such as a handheld device or an embedded controller, which have much smaller computation power and energy supply than the large systems used for training the network. Hence, a method to achieve high prediction accuracy without increasing computation time is needed to enable more applications to be deployed in the real world. To reduce the computation costs in the inference phase, Hinton et al. (2015) created a smaller network for deployment by distilling the knowledge from an ensemble of multiple models. Han et al. (2016) also targeted deployment for small (mobile) devices and showed that large networks can be significantly compressed after training by pruning unimportant connections and by quantizing each connection.

Ensembling multiple predictions is a widely-used technique to improve the accuracy of various machine learning tasks (e.g. Hansen & Salamon (1990), Zhou et al. (2002)) at the cost of more computation power. In the image classification tasks, for example, accuracy is significantly improved by ensembling the local predictions for multiple patches extracted from the input image to make the final prediction. Moreover, accuracy is further improved by using multiple networks trained independently to make local predictions. Krizhevsky et al. (2012) averaged 10 local predictions using 10 patches extracted from the center and the 4 corners with and without horizontal flipping in their Alexnet paper. They also used up to 7 networks and averaged the prediction to get higher accuracy. GoogLeNet by Szegedy et al. (2015) averaged up to 1,008 local predictions by using 144 patches

and 7 networks.In some ensemble methods, meta-learning during the training to learn how to best mix the multiple local predictions from the networks is used (e.g. Tekin et al. (2015)). In the Alexnet or GoogLeNet papers, however, the significant improvements were obtained by just averaging the local predictions without the meta-learning. In this paper, we do not use meta-learning either.

Although the benefits of ensemble prediction are quite significant, one obvious drawback is its higher execution cost during inference. If we make the final prediction by ensembling 100 predictions, we need to make 100 local predictions, and hence the execution cost will be 100 times as high as that without ensembling. This higher execution cost limits the real-world use of ensembling especially on small devices.

In this paper, we first describe our insights on relationship between the probability of the prediction and the effect of ensembling with current deep neural networks; ensembling does not help mispredictions for inputs predicted with a high probability, i.e. the output from the softmax. This finding motivates us to develop an adaptive ensemble prediction, which achieves the benefits of ensembling with much smaller additional costs. During the ensembling process, we calculate the confidence level of the probability obtained from local predictions for each input. If an input reaches a high enough confidence level, we stop ensembling and making more local predictions for this input to avoid wasting computation power. We evaluated the adaptive ensembling by using four image classification datasets: ILSVRC 2012, CIFAR-10, CIFAR-100, and SVHN. Our results showed that the adaptive ensemble prediction reduces the computation cost significantly while achieving similar accuracy to the static ensemble prediction with the fixed number of local predictions. We also showed that our statistically rigorous confidence-level-based termination condition reduces the burden of the task-dependent parameter tuning compared to the naive termination condition based on the predefined threshold in the probability in addition to yielding a better accuracy with the same cost (or lower cost for the same accuracy).

## 2 ENSEMBLING AND PROBABILITY OF PREDICTION

This section describes the observations that have motivated us to develop our proposed technique: how the ensemble prediction improves the accuracy of predictions with different probabilities.

To show the relationship between the probability of the prediction and the effect of ensembling, we evaluate the prediction accuracy for the ILSVRC 2012 dataset with and without ensembling of two predictions made by two independently trained networks. Figure 1(a) shows the results of this experiment with GoogLeNet; the two networks follow the design of GoogLeNet and use exactly the same configurations (hence the differences come only from the random number generator). In the experiment, we 1) evaluated the 50,000 images from the validation set of the ILSVRC 2012 dataset using the first network without ensembling, 2) sorted the images by the probability of the prediction, and 3) evaluated the images with the second network and assessed the accuracy after ensembling two local predictions using the arithmetic mean. The x-axis of Figure 1(a) shows the percentile of the probability from high to low, i.e. going left (right), the first local predictions become more (less) probable. The gray dashed line shows the average probability for each percentile class. Overall, the ensemble improves accuracy well, although we only averaged two predictions. Interestingly, we can observe that the improvements only come in the right of the figure. There are almost no improvements by ensembling two predictions on the left side, i.e. input images with highly probable local predictions, even when there is a non-negligible number of mispredicted inputs. For example, in the 50- to 60-percentile range with GoogLeNet, the top-1 error rate is 29.6% and is not improved by averaging two predictions from different networks.

For more insight into the reason of this characteristics, Figure 2(a) shows the breakdown of 5000 samples in each 10-percentile range into four categolies based on 1) whether the first prediction is correct or not and 2) whether the two network makes the same prediction or different predictions. When a prediction with a high probability is made first, we can observe that another local prediction tends to make the same prediction regardless of its correctness. In the highest 10-percentile range, for instance, two independently trained networks make the same misprediction for all the 43 mispredicted samples. The two networks make different predictions only in two out of 5,000 samples even when we include the correct predictions. In the 10- to 20-percentile range, two networks generate different predictions only in three out of 139 mispredicted samples. Ensembling does not work well when local predictions tend to make the same mispredictions. The insufficient expressiveness in the

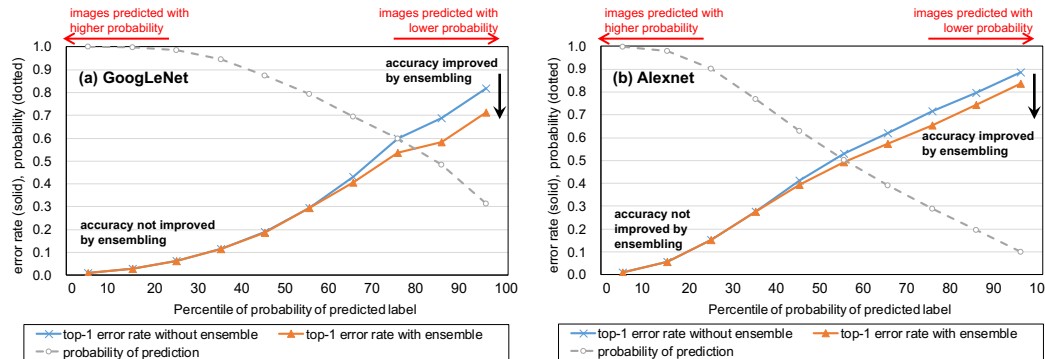

Figure 1: Improvements by ensemble and probabilities of predictions in ILSVRC 2012 validation set. X-axis shows percentile of probability of first local predictions from high (left) to low (right). Ensemble reduces error rates for inputs with low probabilities but does not affect inputs with high probabilities.

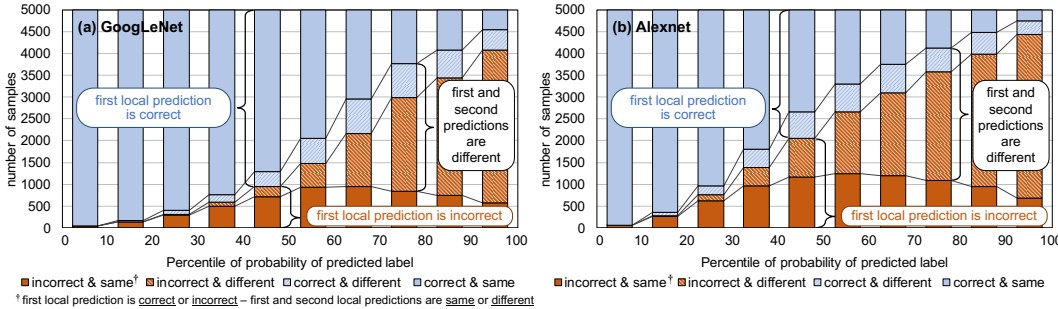

Figure 2: Breakdown of samples into four categories based on 1) whether first local prediction is correct or incorrect and 2) whether first and second predictions are same or different. X-axis shows percentile of probability of first local predictions from high (left) to low (right). Two networks tend to make same (mis)predictions for samples predicted with a high probability (left).

used model or the lack of the appropriate training data may lead to such common mispredictions among local predictions. For many of the mispredictions with low probabilities, the two networks make different predictions and hence the ensemble works for them.

To determine whether or not this characteristic of ensembling is unique to GoogLeNet architecture, we conducted the same experiment using Alexnet as another network architecture and show the results in Figure 1(b) and 2(b). Although the prediction error rate is higher for Alexnet than for GoogLeNet, we observe similar characteristics of improvements by ensembling.These characteristics of the improvements by the ensemble are not unique to an ILSVRC dataset; we have observed similar trends in other datasets.

These results motivate us to make our adaptive ensemble prediction for reducing the additional cost of ensembling while keeping the benefit of improved accuracy. Once we obtain high enough prediction probability for an input image, doing further local predictions and ensembling will waste computation power without improving accuracy. The challenge is how to identify the condition in which to terminate ensembling. As described later, we identify the termination condition based on the confidence level of the probability works well for all the tested datasets.

## 3  RELATED WORK

Various prediction methods that ensemble the outputs from many classifiers (e.g. neural networks) have been widely studied to achieve higher accuracy in machine learning tasks. Boosting (Freund and Schapire 1996) and Bagging (Breiman (1996)) are famous examples of ensemble methods. Boosting and Bagging produce enough variances in classifiers included in an ensemble by chang-

ing the training set for each classifier. In recent studies on image classifications with deep neural networks, however, random numbers (e.g. for initialization or for ordering input images) used in the training phase can give sufficient variances in networks even using the same training set for all classifiers (networks). Hence, we use networks trained using the same training set and network architecture in this study.

The higher execution cost of the ensembling is a known problem, so we are not the first to attack it. For example, Hinton et al. (2015) also tackled the high execution cost of the ensembling. Unlike us, they trained a new smaller network by distilling the knowledge from an ensemble of networks by following Buciluă et al. (2006). To accelerate binary classification tasks, such as face detection, soft-cascade (e.g. Bourdev & Brandt (2005), Zhang & Viola (2008)) is a famous technique. In soft-cascade, multiple weak sub-classifiers are trained to reject a part of the negative inputs. Hence, as the entire strong classifier, many easy-to-reject inputs are rejected in early stages without consuming huge computation time. Unlike soft-cascade, our technique addresses multi-class classification tasks. Also ours is an inference-time technique and does not affect the training phase. Ours can be used even with only one network to make efficient ensemble. However, our basic insight may potentially useful for extending soft-cascade to use in multi-class classification tasks. Another series of studies to accelerate classification tasks with two or few classes is based on dynamic pruning of majority voting (e.g. Hernández-Lobato et al. (2009), Soto et al. (2016)). Like our technique, the dynamic pruning uses a certain confidence level to prune the ensembling with the sequential voting process to avoid wasting the computation time. We show that the confidence-level-based approach is quite effective to accelerate the ensembling by averaging local predictions in many-class classifications tasks with deep neural networks when we use the output of the softmax as the probability of the local predictions. Some existing classifiers with a deep neural network (e.g. Bolukbasi et al. (2017)) take an early exit approach similar to ours. In our study, we study how the exit condition for the termination affects the execution time and the accuracy in detail and showed our confidence-level-based condition works better than the naive threshold-based conditions.

In our technique, we use the probability of the predictions to control the ensembling during the inference. Typically, the probability of the prediction generated by the softmax is used during the training of the network; the cross entropy of the probabilities is often used as the objective function of the optimization. However, using the probability for purposes other than the target of the optimization is not unique to us. For example, Hinton et al. (2015) used the probabilities from the softmax while distilling the knowledge from the ensemble. As far as we know, ours is the first study focusing on the relationship between the probability of the prediction and the effect of ensembling with current deep neural networks.

Opitz & Maclin (1999) showed an important observation related to ours. They showed that the large part of the gain of ensembling came from the ensembling of the first few local predictions. Our observation discussed in the previous section enhances Opitz's observation from a different perspective: most gain of the ensembling comes from inputs with low probabilities in the prediction.

## 4 ADAPTIVE ENSEMBLE PREDICTION

### 4.1 BASIC IDEA

This section details our proposed adaptive ensemble prediction method. As shown in Figure 1, the ensemble typically does not improve the accuracy of predictions if a local prediction is highly probable. Hence, we terminate ensembling without processing all $N$ local predictions on the basis of the probabilities of the predictions. We execute the following steps:

1. start from $i = 1$

2. obtain $i$-th local prediction, i.e. the probability for each class label. We denote the probability for label $L$ of $i$-th local prediction $p_{L,i}$

3. calculate the average probabilities for each class label

$$\langle p_L \rangle_i = \frac{\sum_{j=1}^{i} p_{L,j}}{i} \tag{1}$$

4. if $i < N$ and the termination condition is not satisfied, increment $i$ and repeat from step 2

5. output the class label that has the highest average probability $\arg\max_L (\langle p_L \rangle_i)$ as the final prediction.

## 4.2 CONFIDENCE-LEVEL-BASED TERMINATION CONDITION

For the termination condition in Step 4, we propose a condition based on a confidence level.

We can use a naive condition on the basis of a pre-determined threshold $T$ to terminate the ensembling, i.e. we just compare the highest average probability $\max_L (\langle p_L \rangle_i)$ against the threshold $T$. If the average probability exceeds the threshold, we do not execute further local predictions for ensembling. As we empirically show later, the best threshold $T$ heavily depends on task. To avoid this difficult tuning of the threshold $T$, we propose more statistically rigorous condition in this paper.

Instead of the pre-defined threshold, we can use the confidence intervals (CIs) as a termination condition. We first find the label that has the highest average probability (*predicted label*). Then, we calculate the CI of the probabilities using $i$ local predictions. If the calculated CI of the predicted label does not overlap with the CIs for other labels, i.e. the predicated label is the best prediction with a certain confidence level, we terminate the ensembling and output the predicted label as the final prediction.

We calculate the confidence interval for the probability of label $L$ using $i$ local predictions by

$$\langle p_L \rangle_i \pm z \frac{1}{\sqrt{i}} \sqrt{\frac{\sum_{j=1}^{i} (p_{L,j} - \langle p_L \rangle_i)^2}{i-1}} \tag{2}$$

Here, $z$ is defined such that a random variable $Z$ that follows the Student's-t distribution with $i - 1$ degrees of freedom satisfies the following condition: $Pr\left[Z \leq z\right] = 1 - \alpha/2$. $(1 - \alpha)$ is the confidence level and $\alpha$ is called the significance level. We can read the value $z$ from a precomputated table at runtime. To compute the confidence interval with small number of samples (i.e. local predictions), it is known that the Student's-t distribution is more suitable than the normal distribution. When the number of local predictions increases, the Student's-t distribution approximates the normal distribution.

Preferably, we want to do pair-wise comparisons between the predicted label and all other labels. However, computing CIs for all labels is costly, especially when there are many labels. To avoid excess costs of computing CIs, we compare the probability of the predicted label against the total of the probabilities of other labels. Since the total of the probabilities of all labels (including the predicted label) is 1.0 by definition, the total of the probabilities for the labels other than the predicted label are $1 - \langle p_L \rangle_i$ and the CI is the same size as that of the probability of the predicted label. Hence, our termination condition is

$$\langle p_L \rangle_i - (1 - \langle p_L \rangle_i) > 2z \frac{1}{\sqrt{i}} \sqrt{\frac{\sum_{j=1}^{i} (p_{L,j} - \langle p_L \rangle_i)^2}{i-1}} \tag{3}$$

We avoid computing CI if $\langle p_L \rangle_i < 0.5$ to avoid wasteful computation because the termination condition of equation 2 cannot be met in such cases. Since the CI cannot be calculated with only one local prediction as is obvious from equation (3) to avoid zero divisions, we can use a hybrid of the two termination conditions. We use the static-threshold-based condition only for the first local prediction with a quite conservative threshold (99.99% in the current implementation) to terminate ensembling only for trivial inputs as early as possible, and after the second local prediction is calculated, the confidence-level-based condition of equation (3) is used.

## 5 EXPERIMENTS

### 5.1 IMPLEMENTATION

In this section, we investigate the effects of adaptive ensemble prediction on the prediction accuracy and the execution cost using various image classification tasks: ILSVRC 2012, Street View House Numbers (SVHN), CIFAR-10, and CIFAR-100 (with fine and course labels) datasets.

For the ILSVRC 2012 dataset, we use GoogLeNet as the network architecture and train the network using the stochastic gradient descent with momentum as the optimization method. For other datasets, we use a network that has six convolutional layers with batch normalization (Ioffe & Szegedy (2015)) followed by two fully connected layers. We used the same network architecture except for the number of neurons in the output layer. We train the network using Adam (Kingma & Ba (2014)) as the optimizer. For each task, we trained two networks independently. During the training, we used data augmentations by extracting a patch from a random position of the input image and using random horizontal flipping. Since adaptive ensemble prediction is an inference-time technique, it does not affect the network training. We executed the training and the inference on a Tesla K40 GPU for the ILSVRC 2012 dataset and a Tesla K20 GPU for other datasets.

We averaged up to 20 local predictions using ensembling. We created 10 patches from each input image by extracting from the center and the four corners with and without horizontal flipping by following Alexnet. For each patch, we made two local predictions using two networks. The patch size is 224x224 for the ILSVRC 2012 dataset and 28x28 for the other datasets. We made local predictions in the following order: (center, no flip, network1), (center, no flip, network2), (center, flipped, network1), (center, flipped, network2), (top-left, no flip, network1), ..., (bottom-right, flipped, network2). Since averaging local predictions from different networks typically yield better accuracy and hence we use this order for both our adaptive ensembling and fixed-number static ensembling. For the inference, we use a batch of 200 inputs. As we repeated local predictions, the batch became smaller as computation for parts of inputs terminated.

## 5.2 RESULTS

To study the effects of our adaptive ensemble on the computation cost and accuracy, we show the relationship between them for ILSVRC 2012 and CIFAR-10 datasets in Figure 3. In the figure, the x-axis is the number of ensembled predictions, so smaller means faster. The y-axis is the improvements in classification error rate over the baseline (no ensemble), so higher means better. We evaluated the static ensemble (averaging the fixed number of predictions) by changing the number of predictions to average and our adaptive ensemble. For the adaptive ensemble, we also evaluated with two termination conditions: with naive static threshold and with confidence interval. We tested the static-threshold-based conditions by changing the threshold $T$ and drew lines in the figure. Similarly, we evaluated the confidence-level-based condition with three confidence levels frequently used in statistical testing: 90%, 95% and 99%.

From the figure, there is an obvious tradeoff between the accuracy and the computation cost. The static ensemble with 20 predictions is at one end of the tradeoff because it never terminates early. The baseline, which does not execute ensemble, is at the other end, which always terminates at the first prediction regardless of the probability. Our adaptive ensemble with the confidence-level-based condition achieved better accuracy with the same computation cost (or smaller computation cost for the same accuracy) compared to the static ensemble or the naive adaptive ensemble with a static threshold. The gain with the confidence-level-based condition over the static-threshold-based was significant for CIFAR-10 whereas it was marginal for ILSVRC 2012. These two datasets show the largest and smallest gain with the confidence-level-based condition over the static-threshold-based condition; other datasets showed improvements between those of the two datasets shown in Figure 2.

When comparing two termination conditions in the adaptive ensemble, the confidence-level-based condition eliminates the burden of the parameter tuning compared to the naive threshold-based condition in addition to the benefit of the reduced computation cost. Obviously, how to decide the best threshold $T$ is the most important problem for the static-threshold-based condition. The threshold $T$ can be used as a knob to control the tradeoff between the accuracy and the computation cost, but the static threshold tuning is highly dependent on the dataset and task. From figure 3, for example, $T = 80\%$ seems to be a reasonable choice for ILSVRC 2012, but it is a problematic choice for CIFAR-10. For the confidence-level-based condition, the confidence level also controls the tradeoff. However, the differences in the computation cost and the improvements in accuracy due to the choice of the confidence level were much less significant and less sensitive to the current task than the differences due to the static threshold. Hence task-dependent fine tuning of the confidence level is not as important as the tuning of the static threshold. The easier (or no) tuning of the parameter is an important advantage of the confidence-level-based condition.

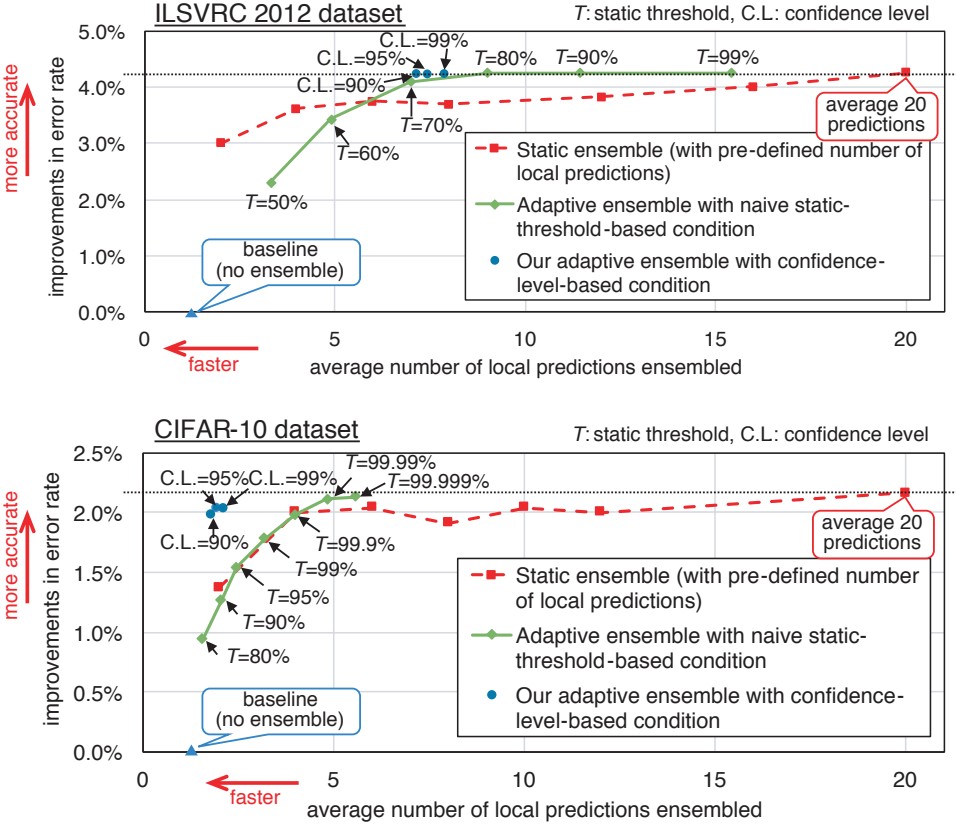

Figure 3: Prediction accuracy and computation cost with static ensemble and our adaptive ensemble using different termination conditions.Confidence-level-based condition achieved better accuracy than static-threshold-based conditions with same computation cost especially for CIFAR-10. Tuning of confidence level (CL) is less sensitive than that of static threshold.

Tables 1, 2, and 3 show how adaptive ensemble prediction affected the accuracy of predictions and the execution costs in more detail for five datasets. Here, for our adaptive ensemble, we use the confidence-level-based termination condition with a 95% confidence level based on the results of Figure 3. We test two different configurations: with one network (i.e. up to 10 local predictions) and two networks (up to 20 local predictions). In all datasets, the ensemble improves the accuracy in a tradeoff for the increased execution costs as expected. Using two networks doubles the number of local predictions on average (from 10 to 20) and increased both the benefit and drawback. If we use further local predictions (e.g. original GoogLeNet averaged up to 1,008 predictions), the benefit and the cost will become much more significant. Comparing our adaptive ensemble with the static ensemble, our adaptive ensemble similarly improves accuracy (from 92% to 99% when we use two networks and from 83% to 99% when we use one network) while reducing the number of local predictions used in the ensembles; the reductions are up to 6.9x and 12.7x for the one-network and two-network configurations. The reduced numbers of local predictions result in shorter execution time; the speedup was by 2.1x to 2.8x and by 2.3x to 3.5x for the one-network and two-network configurations, respectively. The reductions in the execution time over the static ensemble are smaller than the reduction in the number of averaged predictions because of the additional overhead due to the confidence interval calculation, which is written in Python in the current implementation. Also, mini batches gradually become small as ensembling for parts of inputs terminated. The smaller batch sizes reduce the efficiency of execution on current GPUs. Since the speedup by our adaptive technique over the static ensemble becomes larger as the number of max predictions to ensemble increases, the benefit of our adaptive technique will become more impressive if we use larger ensemble configurations.

Table 1: Prediction accuracy with and without adaptive ensemble

| dataset | | # class labels | classification error rate (lower is better) with one network | | | classification error rate with two networks | |
|---|---|---|---|---|---|---|---|
| | | | no ensemble | naive ensemble | our adaptive ensemble | naive ensemble | our adaptive ensemble |
| CIFAR-10 | | 10 | 8.39% | 6.97% (-1.41%) | 7.00% (-1.39%) | 6.23% (-2.16%) | 6.34% (-2.04%) |
| SVHN | | 10 | 4.40% | 3.44% (-0.96%) | 3.50% (-0.90%) | 3.19% (-1.21%) | 3.29% (-1.11%) |
| CIFAR-100 (course label) | | 20 | 20.63% | 17.84% (-2.79%) | 18.04% (-2.59%) | 16.56% (-4.07%) | 16.78% (-4.06%) |
| CIFAR-100 (fine label) | | 100 | 30.28% | 27.04% (-3.24%) | 27.34% (-2.94%) | 25.04% (-5.24%) | 25.15% (-5.13%) |
| ILSVRC 2012 | top-1 error | 1000 | 32.36% | 30.21% (-2.15%) | 30.26% (-2.10%) | 28.11% (-4.25%) | 28.12% (-4.24%) |
| | top-5 error | | 12.67% | 11.11% (-1.37%) | 11.35% (-1.14%) | 9.99% (-2.50%) | 10.21% (-2.28%) |

Ratios in parenthesis show improvements in error rate over baseline (no ensemble).

Table 2: Number of local predictions ensembled with and without adaptive ensemble

| dataset | # local predictions ensembled (lower is better) with one network | | | # local predictions ensembled with two networks | |
|---|---|---|---|---|---|
| | no ensemble | naive ensemble | our adaptive ensemble | naive ensemble | our adaptive ensemble |
| CIFAR-10 | | | 1.66 | | 1.92 |
| SVHN | | | 1.44 | | 1.57 |
| CIFAR-100 c | 1 | 10 | 2.74 | 20 | 4.09 |
| CIFAR-100 f | | | 3.59 | | 5.93 |
| ILSVRC 2012 | | | 3.94 | | 7.40 |

Table 3: Execution time with and without adaptive ensemble

| dataset | execution time per sample (lower is better) with one network | | | execution time per sample with two networks | |
|---|---|---|---|---|---|
| | no ensemble | naive ensemble | our adaptive ensemble | naive ensemble | our adaptive ensemble |
| CIFAR-10 | 0.30 msec (1.0x) | 2.55 msec (8.37x) | 0.98 msec (3.20x) | 4.98 msec (16.34x) | 1.61 msec (5.28x) |
| SVHN | 0.28 msec (1.0x) | 2.52 msec (9.09x) | 0.89 msec (3.20x) | 4.96 msec (17.83x) | 1.43 msec (5.13x) |
| CIFAR-100 (course label) | 0.31 msec (1.0x) | 2.55 msec (8.28x) | 1.04 msec (3.58x) | 4.99 msec (16.16x) | 1.87 msec (6.01x) |
| CIFAR-100 (fine label) | 0.31 msec (1.0x) | 2.56 msec (8.36x) | 1.25 msec (4.07x) | 4.99 msec (16.28x) | 2.15 msec (7.03x) |
| ILSVRC 2012 | 3.75 msec (1.0x) | 35.84 msec (9.56x) | 16.67 msec (4.45x) | 70.74 msec (18.86x) | 30.10 msec (8.03x) |

Ratios in parenthesis show relative slowdown over baseline (no ensemble).

## 6 CONCLUSION

In this paper, we described our adaptive ensemble prediction to reduce the computation cost of ensembling many predictions. We were motivated to develop this technique by our observation that ensembling does not improve the prediction accuracy if predictions are highly probable. Our experiments using various image classification tasks showed that our adaptive ensemble makes it possible to avoid wasting computing power without significantly sacrificing the prediction accuracy by terminating ensembles based of the probabilities of the local predictions. The benefit of our technique will become larger if we use more predictions in an ensemble. Hence, we expect our technique to make the ensemble techniques more valuable for real-world systems by reducing the total computation power required while maintaining good accuracies and throughputs.

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
