# OpenReview forum: "Fast and Accurate Inference with Adaptive Ensemble Prediction for Deep Networks"
_ICLR.cc/2018/Conference — Reject_

### Official Review · AnonReviewer1 · 2017-11-24
**A simple way of improving the prediction speed of ensembles of probabilistic predictors.**

**Rating:** 6
**Confidence:** 3

**Review:**

In this paper it is described a method that can be used to speed up the prediction process of ensembles of classifiers that output probabilistic predictions. The method proposed is very simple and it is based on the observation that in the case that the individual predictors are very sure about the potential class label, ensembling many predictions is not particularly useful. It seems it is most useful when the individual classifier are most unsure, as measured by the output probabilities. The idea proposed by the authors is to compute an estimate of the probability that the class with the highest probability will not change after querying more predictors from the ensemble. This estimate is obtained by using a t-student distribution for the distribution of the average maximum probability.

The paper is generally well written with a few mistakes that can be easily corrected using any spell checking tool.

The experiments carried out by the authors are convincing. It seems that their proposed approach can speed up the predictions of the ensemble by an important factor. The benefits of using ensemble methods are also evident, since they always improve the performance of a single classifier.

As far as I know this work is original. However, it is true that several similar ensemble pruning techniques exist for multi-class problems in which one uses majority voting for computing the combined prediction of the ensemble. Therefore it is unclear what are the advantages of the proposed method with respect to those ones. This is, in my opinion, the weakest point of the paper.

---

### Official Review · AnonReviewer2 · 2017-11-28
**Good idea but poor execution**

**Rating:** 5
**Confidence:** 4

**Review:**

The authors propose and evaluate an adaptive ensembling threshold using estimated confidence intervals from the t-distribution, rather than a static confidence level threshold. They show it can provide significant improvements in accuracy at the same cost as a naive threshold.

This paper has a nice simple idea at its core, but I don't think it's fully developed. There's a few major conceptual issues going on:

- The authors propose equation (3) as a stopping criterion because "computing CIs for all labels is costly." I don't see how this is true in any sense. The CI computation is literally just averages of a few numbers, which should be way less than the massive matrix multiplies needed to *generate* those numbers in the neural network. Computing pair-wise comparisons naively in O(n^2) time could potentially blow up if the number of output labels is massive, but then you should still be able to keep some running statistics to avoid having to do a quadratic number of comparisons (e.g. the threshold is just the highest bound of any CI you encounter, so you keep track of both the max predicted confidence and max CI so far...then you have your answer in O(n) time.) I think the real issue is that the authors state that the confidence interval computation code is written in Python. That is a huge knock against this paper: When writing a paper about inference time, it's just due diligence to do the most basic inference time optimizations (such as implementing an operation which should be effectively free in a C++ plugin.)

- So by using (3) instead of the original proposed CI comparison that motivated this approach, the authors require that the predicted probability be greater than 1/2 + the CI at the given alpha level. This means that for problems with very large output spaces, getting enough probability mass to get over that 1/2 absolute threshold is potentially going to require a minimum number of evaluations and put a cap on the efficiency gain. This is what we see in Figure 3: for the few points evaluated, when the output space is large (ILSVRC 2012) there is no effective difference between the proposed method and a static threshold of 70%, indicating that the CI of 90% is roughly working out to be the 50% minimum + ~20% threshold from the CI.

- Thus the experiments in this paper don't really add much value in understanding the benefits of this approach as currently written. For due diligence, there should be the following:

1. Show the distribution of computing thresholds from the CI.  Then compute, for a CI of 0.8, 0.9, etc., what is the effective threshold on average? Then for every *average threshold* from the CI method, apply that as a static threshold. Then you will get exactly the delta of your method over the static threshold method.

2. Do the same, but using the pairwise CI comparison method.

3. The same again, but now show how effective this is as a function of the size of the output label space.  E.g. add these numbers to Table 1 and Table 2 (for every "our adaptive ensemble", put the equivalent static threshold.)

4. Implement the CI computation efficiently if you are going to report actual runtimes. Note that for a paper like this, I don't think the runtimes are as important as the # of evaluations in the ensemble, so this is less important.

- With the above experiments I think this would be a good paper.

---

### Official Review · AnonReviewer3 · 2017-11-29
**nice and simple idea; not sure of its real impacts**

**Rating:** 5
**Confidence:** 4

**Review:**

Summary

The authors argue that ensemble prediction takes too much computation time and resource, especially in the case of deep neural networks. They then address the problem by proposing an adaptive prediction approach. The approach is based on the observation that it is most important for ensemble approaches to focus on the "uncertain" examples. The proposed approach thus conducts early-stopping prediction when the confidence (certainty) of the prediction is high enough, where the confidence is based on the confidence intervals of (multi-class) labels based on the student-t distribution. Experiments on vision datasets demonstrate that the proposed approach is effective in reducing computation resources while maintaining sufficient accuracy.

Comments

* The experiments are limited in the scope of (image) multi-class classification. It is not clear whether the proposed approach is effective for other classification tasks, or even more sophisticated tasks like multi-label classification or sequence tagging.
* The idea appears elegant but rather straightforward. One important baseline that is easy but not discussed is to set a static threshold on pairwise comparison (p_max - p_secondmax). Would this baseline be competitive with the proposed approach? Such a comparison is able to demonstrate the benefits of using confidence interval.
* The overall improvement in computation time seems to be within a constant scale, which can be easily achieved by doing ensemble prediction in parallel (note that the proposed approach would require predicting sequentially). So are there real applications that can benefit from the improvement?
* typo: p4, line19, neural "netowkrs" -> neural "networks"

---

### Author Response · Authors · 2018-01-05
**Author response**

First of all, we like to thank the reviewers for their valuable comments.

As reviewers pointed out, there can be many different approaches for the termination condition (e.g. using second max, or using pairwise CI comparison).
We intend to claim that our CI-based termination is a reasonable approach with robustness and accuracy compared to the naive static threshold condition.
However, I do not intend to claim that it is really the best approach among all possible termination conditions; testing wider range of the termination conditions to identify the best approach will be a future work.

I believe that the most important contribution of this paper is findings in Section 2; the ensembling does not help samples with high probability and hence the probability can be used to adaptively control the ensemble if we use a reasonable threshold.
To emphasize this point, I touched up abstract and introduction (as well as fixing typos) and updated the submission.

---

> ### Comment · AnonReviewer2 · 2018-01-10
> **re. response**
>
> I don't think that observation warrants a full paper. You have to do something useful with it to have impact. As a reviewer, it's very frustrating to hear that you want to leave development of your observation for future papers. It's not enough to just point out something interesting. Interesting ideas are cheap and easy to come by; making them useful for someone else in the community is hard.
>
> My point in the review was that I don't think you've shown enough of this impact to warrant acceptance. For example, your experiments show that your approximation doesn't improve over a static threshold for the large output space example. My comment was that (1) I think this is most likely due to your approximation and (2) there is no reason to use an approximation in the first place because full pairwise shouldn't be costly if you implement it correctly. Please correct me if I'm wrong about #2, but even so, you need to show #1 to provide some context to your results.
>
> To be clear, I never evaluated the claim that "this is the best among all possible methods." I evaluated "this paper explores the proposed idea (ensembling with dynamic thresholds) fully enough to be useful to the community on its own."
>
> So, I apologize but my review stands -- I appreciate that you took the time to read and respond to the reviews, but given your response I still do not think this paper should be accepted.

---

> > ### Author Response · Authors · 2018-01-10
> > **Re: re. response**
> >
> > Thank you so much for the clarification. I understand what you intend in your review.

---

### Decision · Program_Chairs · 2018-01-29
**ICLR 2018 Conference Acceptance Decision**

**Decision:**

Reject

**Comment:**

The manuscript proposes a simple technique for adaptive ensemble prediction. Unfortunately, several significant concerns were raised (by R2 and R3) that this AC agrees with. Both R2 and R3 asked fairly specific questions and requested follow-up experiments, which have not been addressed.